# The Association between Gut Microbiota and Uremia of Chronic Kidney Disease

**DOI:** 10.3390/microorganisms8060907

**Published:** 2020-06-16

**Authors:** Ji Eun Kim, Hyo-Eun Kim, Ji In Park, Hyunjeong Cho, Min-Jung Kwak, Byung-Yong Kim, Seung Hee Yang, Jung Pyo Lee, Dong Ki Kim, Kwon Wook Joo, Yon Su Kim, Bong-Soo Kim, Hajeong Lee

**Affiliations:** 1Department of Internal Medicine, Seoul National University Hospital, Seoul 03080, Korea; beeswaxag@naver.com (J.E.K.); dkkim73@gmail.com (D.K.K.); junephro@gmail.com (K.W.J.); yonsukim@snu.ac.kr (Y.S.K.); 2Department of Internal Medicine, Korea University Guro Hospital, Seoul 08308, Korea; 3Seoul National University Hospital Biomedical Research Institute, Seoul 03082, Korea; hyoeun9492@gmail.com; 4Department of Internal Medicine, Kangwon National University Hospital, Chuncheon 24295, Korea; park.jiin@gmail.com; 5Department of Internal Medicine, Chungbuk National University Hospital, Cheongju 28644, Korea; mdhjcho@gmail.com; 6ChunLab, Inc., Seoul 06725, Korea; minjung.kwak@chunlab.com (M.-J.K.); bykim@chunlab.com (B.-Y.K.); 7Kidney Research Institute, Seoul National University, Seoul 08826, Korea; ysh5794@gmail.com (S.H.Y.); nephrolee@gmail.com (J.P.L.); 8Department of Internal Medicine, Seoul National University Boramae Medical Center, Seoul 07061, Korea; 9Department of Life Science, Multidisciplinary Genome Institute, Hallym University, Chuncheon 24252, Korea

**Keywords:** chronic kidney disease, gut microbiota, uremic toxin, metabolism, *Oscillibacter*

## Abstract

Chronic kidney disease (CKD)-associated uremia aggravates—and is aggravated by—gut dysbiosis. However, the correlation between CKD severity and gut microbiota and/or their uremic metabolites is unclear. We enrolled 103 CKD patients with stage 1 to 5 and 46 healthy controls. We analyzed patients’ gut microbiota by MiSeq system and measured the serum concentrations of four uremic metabolites (*p*-cresyl sulfate, indoxyl sulfate, *p*-cresyl glucuronide, and trimethylamine *N*-oxide) by liquid chromatography–tandem mass spectrometry. Serum concentrations of the uremic metabolites increased with kidney function deterioration. Gut microbial diversity did not differ among the examined patient and control groups. In moderate or higher stage CKD groups, *Oscillibacter* showed positive interactions with other microbiota, and the proportions of *Oscillibacter* were positively correlated with those of the uremic metabolites. The gut microbiota, particularly *Oscillibacter*, was predicted to contribute to pyruvate metabolism which increased with CKD progression. Relative abundance of *Oscillibacter* was significantly associated with both serum uremic metabolite levels and kidney function. Predicted functional analysis suggested that kidney-function-associated changes in the contribution of *Oscillibacter* to pyruvate metabolism in CKD may greatly affect the gut environment according to kidney function, resulting in dysbiosis concomitant with uremic toxin production. The gut microbiota could be associated with uremia progression in CKD. These results may provide basis for further metagenomics analysis of kidney diseases.

## 1. Introduction

More than one trillion microbes are found in the human body [1], and they encode 3.3 million genes, which is 150-fold higher than the number of genes encoded by the human genome [2]. By creating a stable environment for symbiotic interactions with the host, these microbes play critical metabolic roles in maintaining host health and homeostasis [2,3,4,5]. In chronic kidney disease (CKD), however, the influx of urea, uric acid, and oxalate from the gastrointestinal tract increases as kidney function deteriorates, which can perturb the balance between commensals and pathogens and lead to gut dysbiosis [2,6]. In parallel, the proliferation of urease-producing bacteria may result in the loss of intestinal barrier integrity and a sequential increase of bacterial translocation to the systemic circulation, leading to the deterioration of kidney function [2,7,8].

Gut dysbiosis due to kidney impairment is specifically associated with an alteration in uremic toxin metabolism [9,10]. Various uremic toxins, including p-cresol and indole derivatives, produced by dietary protein metabolism in humans have been identified [11]. The levels of p-cresol and indole derivatives in serum are negatively correlated with kidney function and these protein-bound solutes are inefficiently eliminated by dialysis [12,13,14]. They are also associated with decreased kidney function, cardiovascular disease, and mortality in patients with pre-dialytic CKD [11,12,15,16]. In addition to such amino acid-derived metabolites, trimethylamine-N-oxide (TMAO), which is produced by the microbial degradation of dietary quaternary amines, including choline, betaine, and carnitine, is an important uremic toxin related to the progression of CKD, atherosclerosis, heart attack, and metabolic syndrome [17,18,19].

Although various studies have reported the relationship between gut dysbiosis and kidney function impairment, as well as the involvement of gut microbiota in uremic toxin metabolism [8,19,20], few studies have examined the consequential linkages among gut microbiota, uremic toxins, and kidney dysfunction. This study was conducted to analyze the interrelationships among the gut microbiota, uremic metabolites known to be produced in the gut, and kidney function impairment of patients in various stages of CKD.

## 2. Materials and Methods

### 2.1. Study Participants

This study was approved by the Medical Ethics Committee of Seoul National University Hospital (IRB number: 1808-153-967) and complied with the Declaration of Helsinki. The study included only participants who provided informed consent and agreed to submit fecal specimens to the human stool repository (IRB number: 1802-062-921). The human stool repository includes samples collected from patients who underwent percutaneous kidney biopsy (IRB number: 1508-046-694) and from kidney transplant donors and recipients prior to transplantation (IRB number: 1703-062-839). We also obtained serum samples at the same time as feces collection from registered participants. The serum samples for this study were provided by the Seoul National University Hospital Human Biobank, a member of the National Biobank of Korea. All samples derived from the National Biobank of Korea were obtained with informed consent under institutional review-board-approved protocols.

### 2.2. Clinical Information on Study Participants

We collected demographic information from the study participants, including details on age, sex, height, and weight, and established whether any of these individuals had comorbidities, including hypertension and diabetes mellitus, as determined from their clinical and medication histories compiled in an electronic medical record system. Participants who had used antibiotics within one month were excluded. Furthermore, given that gut microbiota may be affected by the systemic inflammation status, we also collected data on the serum levels of highly sensitive C-reactive protein, which is used as a marker of inflammation.

Kidney function was assessed by measuring serum blood urea nitrogen and creatinine levels at the time of fecal sample collection. The presence of hematuria and amount of proteinuria were quantified based on urine microscopic examination and random urine protein to creatinine ratio determinations, respectively. Plasma hemoglobin and serum albumin concentrations were examined to evaluate the anemia and nutritional status, respectively. The etiologies of CKD were established by pathological confirmation, imaging studies, or clinical diagnosis. The estimated glomerular filtration rate was calculated using the Chronic Kidney Disease Epidemiology Collaboration calculation formula according to CKD staging.

We divided the study participants into the following groups: kidney donors without evidence of kidney disease as healthy controls; patients with stage 1 and 2 CKD as mild CKD; patients with stage 3 and 4 CKD, and stage 5 CKD without dialysis, as moderate to severe CKD; and patients with stage 5 CKD requiring dialysis as end-stage renal disease (ESRD).

### 2.3. Stool DNA Extraction and MiSeq Sequencing

The stool samples collected from all study participants were immediately stored in a deep freezer at −80 °C. Stool DNA extraction was performed using a QIAamp^®^ Fast DNA Stool Mini Kit (Qiagen, Hilden, Germany) according to the manufacturer’s instructions [21]. The extracted DNA was used to amplify the V4–5 variable regions of the 16S rRNA gene. Amplification was performed in accordance with the MiSeq system protocol for preparing a 16S metagenomics sequencing library (Illumina, Inc., San Diego, CA, USA). The amplicons of each sample were purified using Agencourt AMPure XP beads (Beckman Coulter, Brea, CA, USA), and the purified amplicons were quantified using a PicoGreen dsDNA Assay kit (Invitrogen, Carlsbad, CA, USA). Equimolar concentrations of each library were pooled and sequenced using the Illumina MiSeq system (250-base pair paired ends) according to the manufacturer’s instructions.

### 2.4. Sequence Data Analysis

For microbiota analysis, the obtained sequence reads were analyzed using the Microbial Genomics Module of CLC genomic workbench v. 11.0.1 (Qiagen, Aarhus, Denmark). Briefly, raw sequences were merged, and sequences with short read lengths (<400 base pairs of merged reads) or low-quality score and chimeric reads were removed using the USEARCH pipeline v. 11.0.667 (http://www.drive5.com/usearch). Primer sequences were removed from the merged sequences, and filtered sequences were subsequently clustered into operational taxonomic units (OTUs) based on 97% sequence identity. The taxonomic positions of representative sequences in each OTU were assigned based on comparisons with the EzTaxon-e reference database [22]. To compare diversity indices among samples, sequence read numbers were normalized by random subsampling and the indices were calculated using Mothur [23]. Principal coordinate analysis (PCoA) based on Bray–Curtis distances was performed using Calypso to compare microbiota compositions among samples [24]. The functional roles of microbiota were predicted using Phylogenetic Investigation of Communities by Reconstruction of Unobserved States (PICRUSt) [25], and the co-occurrence networks for microbiota in each group were inferred based on Spearman correlation matrices and selected according to *q* < 0.05 (Benjamini–Hochberg-corrected). Networks were constructed for significantly different direct interactions among genera in the different patient groups, with the visualization of networks and calculations performed using CoNet with Cytoscape (v. 3.4.0).

### 2.5. Serum Metabolite Analysis

Serum specimens collected from all study participants were stored in a −180 °C nitrogen tank. To measure metabolites that are representative of uremic toxins, we selected as target metabolites *p*-cresyl sulfate, *p*-cresyl glucuronide, indoxyl sulfate, and TMAO based on the findings of previous studies [11,26,27]. The serum concentrations of these four metabolites were determined by liquid chromatography–tandem mass spectrometry, as described previously [28,29]. Twenty-microliter samples were placed in microtubes, followed by the addition of 20 μL of acetonitrile containing an internal standard and 500 μL of acetonitrile containing 0.1% formic acid. These sample mixtures were vortexed for 30 s, followed by centrifugation at 13,000 rpm for 5 min. The resulting supernatants were transferred into injection vials and subjected to liquid chromatography–tandem mass spectrometry. We used an Agilent 1260 Infinity high-performance liquid chromatography system in conjunction with an API 4000 QTRAP mass spectrometry system (Agilent Technologies, Santa Clara, CA, USA). From the chromatograms, we calculated the area ratios of *p*-cresyl sulfate, *p*-cresyl glucuronide, indoxyl sulfate, TMAO, and the internal standard and determined the concentrations of the metabolites using previously prepared calibration curves. For *p*-cresyl glucuronide, the minimum detection limit was 5 ng/mL, and values were designated as zero when measurements were below this value.

### 2.6. Statistical Analysis

For the baseline characteristics, continuous variables are expressed as the means and standard deviations and categorical variables are expressed as percentages. Differences between samples were evaluated using the Mann–Whitney U and Kruskal–Wallis tests. The *p*-value for trends was calculated using the Stata module “nptrend,” which is an extension of the Mann–Whitney U test that can be used to perform nonparametric tests for trends across ordered groups; *p* < 0.05 was considered to indicate statistical significance. Correlations between microbiota and uremic toxins were determined by linear regression. The Benjamini–Hochberg false discovery rate (FDR) was applied to correct for multiple testing and FDR-adjusted *p*-values of less than 0.05 were considered as significant. Statistical analyses were performed using R v. 3.5.0 (R Core Team), Stata v. 15.1 (StataCorp, College Station, TX, USA), and GraphPad Prism v. 8.1.1 (GraphPad, Inc., San Diego, CA, USA).

## 3. Results

### 3.1. Comparisons of Baseline Characteristics and Serum Uremic Metabolites according to CKD Group

We analyzed samples collected from a total of 149 participants, among whom there were 46 controls and 103 patients with CKD. The patients with CKD comprised 36 subjects with mild CKD, 32 with moderate to severe CKD, and 35 with dialysis-dependent ESRD. The baseline characteristics, laboratory data, and uremic metabolite concentrations were compared among groups (Table 1). We found no significant differences among groups with respect to the distributions of age and sex. Patients with ESRD were, however, significantly leaner than patients in the control and other CKD groups (*p* = 0.003). The participants with lower kidney function had more comorbidities, including hypertension and diabetes mellitus (*p* < 0.001 and 0.020, respectively). In line with expectations, we observed that plasma hemoglobin and serum albumin levels were negatively correlated with CKD severity (*p* < 0.001, in both cases). The proportion of patients with anemia increased significantly with worsening CKD severity (*p* < 0.001). However, serum levels of highly sensitive C-reactive protein did not differ significantly among the four groups.

Measurements of the four uremic metabolites *p*-cresyl sulfate, *p*-cresyl glucuronide, indoxyl sulfate, and TMAO in the sera revealed clear positive correlations with CKD severity (Kruskal–Wallis, *p* < 0.001), although there were no significant differences in the levels of these metabolites between participants in the control and mild CKD groups (Appendix A).

### 3.2. Differences in Microbiota Composition according to CKD Group

A total of 1,796 OTUs were detected in all patients. To compare the diversity indices among samples, read numbers were normalized to 6,400 by random subsampling, and we accordingly detected no significant differences in bacterial diversity among the different patient groups (*p* > 0.05; Figure 1a). However, we detected a larger number of observed OTUs in the ESRD group compared to the control and mild CKD groups, respectively (*p* < 0.05; Figure 1b). PCoA based on Bray–Curtis distances did not clearly distinguish differences in the microbiota in different CKD groups (Figure 1c). There were also no differences among the groups with respect to the proportions of the different phyla, although *Bacteroidetes*, *Firmicutes*, and *Proteobacteria* were dominant phyla in the gut microbiota of all groups (Figure 1d). A comparison of the microbiota among groups at the genus level (Figure 1e) revealed 18 main genera, defined as those constituting more than 1% of the total microbiota in samples. In most samples, *Bacteroides* and *Prevotella* were the dominant genera (with averages of 28.5% and 15.0% in samples, respectively). When we evaluated the differences in genera among groups based on multiple group comparisons, the proportions of *Alistipes*, *Oscillibacter*, *Lachnospira*, *Veillonella*, and *Dialister* were shown to be significantly different among the four groups (*p* < 0.05; Figure 2). The proportions of these genera in the microbiota of the control group were significantly different from those in the moderate to severe CKD group, but not from those in the mild CKD group. Furthermore, the proportions of *Alistipes*, *Oscillibacter*, *Lachnospira*, and *Veillonella* in the mild CKD group differed from those in the moderate to severe CKD group. In contrast, we detected no significant differences in the proportions of *Alistipes*, *Oscillibacter*, *Lachnospira*, *Veillonella*, and *Dialister* in the moderate to severe CKD and ESRD groups. The relative abundance of *Alistipes* and *Oscillibacter* was increased with the progression in CKD severity (tests for trends, *p* = 0.001 and 0.016, respectively), whereas the abundance of *Lachnospira*, *Veillonella*, and *Dialister* decreased with increasing CKD severity (tests for trends, *p* = 0.019, 0.012, and *p* < 0.001, respectively).

To determine whether mutual interactions among the aforementioned five genera and other gut microbes differed according to CKD severity, we performed co-occurrence network analysis. We accordingly found that as kidney function decreased, the network appeared to become more active and complex in terms of both positive and negative interactions of gut microbiota in these five genera with other microbes (Figure 3). *Oscillibacter* and *Veillonella* were found to show the highest values of betweenness centrality in networks and appeared to act as hubs in the microbial networks constructed for patients with moderate or higher stage CKD. *Oscillibacter* showed positive correlations with other genera, whereas *Veillonella* species were negatively correlated. These results indicate that the relative abundance of *Oscillibacter* associated with kidney function impairment gives rise to the co-occurrence of other gut microbiota.

### 3.3. Microbiota-Related Uremic Toxins

We subsequently analyzed differences in the relative abundance of gut microbiota according to the levels of uremic toxins in the sera of patients and controls. The genera found to be associated with the levels of uremic toxins are shown on a logarithmic scale in Table 2. According to the FDR-adjusted *p*-values (*q*-values) in multivariable linear regression, *p*-cresyl sulfate showed a significant association with six major genera, with positive correlations observed with *Alistipes*, *Oscillibacter*, and *Subdoligranulum* (*q* < 0.001, *q* < 0.001, and *q* = 0.023, respectively) and negative correlations with *Lachnospira*, *Veillonella*, and *Megamonas* (*q* = 0.039, *q* = 0.014, and *q* = 0.034, respectively). Serum *p*-cresyl glucuronide and indoxyl sulfate levels were positively correlated with *Alistipes* (*q* = 0.010 and *q* = 0.035, respectively) and *Oscillibacter* (*q* = 0.001 and *q* = 0.037, respectively). Serum TMAO levels were associated only with *Oscillibacter* (*q* = 0.006). Unexpectedly, we found *Oscillibacter* to be associated with all four measured uremic toxins, although each of these metabolites is derived from different parent compounds.

### 3.4. Predicted Functional Analysis of Gut Microbiota among the CKD Groups

The functions of the gut microbiota were predicted based on PICRUSt analysis and a comparison of pathways based on the Kyoto Encyclopedia of Genes and Genomes (KEGG) categories among the different CKD groups. At KEGG Ortholog (KO) level 3, four pathways were predicted to be significantly different among groups (*q* < 0.05; Appendix A). Among these significant pathways, KEGG terms relating to “Pyruvate metabolism” and “Methane metabolism” were predicted to increase with decreasing kidney function, whereas “Riboflavin metabolism” was predicted to decrease with kidney function deterioration (Figure 4a).

Given that pyruvate metabolism represents a key intersection in the network of various metabolic pathways, we focused on this pathway and examined the contribution of major genera to pyruvate metabolism (Figure 4b). We accordingly found that only the proportional contributions of *Oscillibacter* and *Veillonella* showed significant differences among the patient groups (*p* = 0.0011 and *p* = 0.0014, respectively; Figure 4c), and as CKD severity increased, the contribution of *Oscillibacter* to pyruvate metabolism increased, whereas that of *Veillonella* decreased.

As *Oscillibacter* showed some evidence of an association with uremic metabolites and CKD severity as well as differences among CKD groups regarding the proportional contribution to pyruvate metabolism, we performed a detailed analysis of the contribution of *Oscillibacter* to each orthologous gene involved in pyruvate metabolism. Among all orthologs, we found that functional orthologs of the E1 component of pyruvate dehydrogenase (K00161, K00162) showed the highest proportional contribution of *Oscillibacter* (33.0% for K00161 and 33.1% for K00162). Among all genera, *Oscillibacter* also showed a high proportional contribution (21.3%) to the pyruvate dehydrogenase E2 component (K00627). These findings indicate that *Oscillibacter* is a primary source of pyruvate dehydrogenase, a key enzyme involved in glucose oxidation that converts pyruvate to acetyl-CoA. The proportional contributions of *Oscillibacter* to each orthologous gene involved in pyruvate metabolism compared with those of other genera are shown in Appendix A.

Although comparisons among the different patient groups revealed no significant differences in the contributions of *Oscillibacter* to each of pyruvate metabolism orthologs according to CKD severity, we found that the contribution to pyruvate dehydrogenase E1 component decreased by approximately 10% in CKD groups compared with the control group (for K00161: 40.1% in the control, 27.0% in mild CKD, 32.2% in moderate to severe CKD, and 31.4% in ESRD; for K00162: 40.2% in the control, 27.0% in mild CKD, 32.2% in moderate to severe CKD, and 31.4% in ESRD). In contrast, we found that the contribution of *Oscillibacter* to lactate dehydrogenase, which converts pyruvate to lactate, was significantly increased (Kruskal–Wallis *p* = 0.003; *q*-value between control and ESRD = 0.009; Figure 5). In addition, we observed significant increases in the contributions to other collateral metabolic pathways, among which was an increase in oxaloacetate and formate production at the expense of a direct generation of acetyl-CoA via pyruvate dehydrogenase (K01958: pyruvate carboxylase, *p* < 0.001; K00656: formate C-acetyltransferase, *p* = 0.009; Appendix A and Appendix A).

## 4. Discussion

The gut microbiota as a source of uremic toxin accumulation in patients with CKD has gained attention. CKD affects bacterial fermentation processes, including colonic transit time and colon microbiota composition, resulting in changes in the microenvironment of the colon [2,6]. However, the integrating effect of CKD on the gut microbiota and their associated metabolites have not been characterized in detail. In this study, we investigated the associations between specific gut microbes, including *Oscillibacter*, and levels of uremic metabolites, along with changes in predicted metabolic pathways related to kidney dysfunction in patients with CKD.

Although several studies have analyzed the gut microbiota in patients with CKD, most studies focused on the differences between patients with ESRD and healthy controls. For example, Vaziri et al. identified a significant elevation in the relative abundance of 190 OTUs in patients with ESRD compared with healthy controls [30]. Similarly, using a bacterial culture method based on diluted stool samples, Hida et al. observed elevated contents of several gut bacteria in patients on hemodialysis compared to healthy controls [31]. In contrast to these previous studies, we focused on patients with pre-dialytic CKD, who were subclassified based on their estimated glomerular filtration rate to examine serial changes in gut microbiota according to kidney functional impairment. We accordingly identified specific gut microbes (*Alistipes*, *Oscillibacter*, *Lachnospira*, *Veillonella*, and *Dialister*), the relative abundances of which showed continuous change concomitant with changes in kidney function. Given that previous studies only investigated differences in OTUs at the family level or differences in OTUs between control and subjected with ESRD [30,31], it is difficult to properly compare the changes in OTUs discovered in previous studies and those in the current study. However, in the study of Varizi et al. [30], the class *Clostridia* which includes *Oscillibacter*, was reported to show a significant elevation in patients with ESRD compared with control subjects, thereby indicating that our findings are at least partially consistent with those reported previously.

Of the bacterial taxa showing significant differences among the CKD groups, *Oscillibacter* and *Veillonella* appear to have important roles as network hubs for the other microbes in the analysis of network co-occurrence. Particularly, *Oscillibacter* showed significant positive associations with diverse genera in patients with advanced CKD, indicating that the bacterial species in this genus interact with those in other bacteria in the gut microbiota of patients with advanced CKD. Moreover, we found that *Oscillibacter* showed common associations with the four uremic metabolites *p*-cresyl sulfate, *p*-cresyl glucuronate, indoxyl sulfate, and TMAO, which are derived from a diverse range of parent compounds, including phenols, indoles, and quaternary amines. These broad effects on uremic metabolites and other microbes suggest that *Oscillibacter* play a pivotal role in creating a favorable inflammatory environment that facilitates the proliferation and activation of a number of uremic toxin-producing pathogens. Although they are poorly represented in culture collections, the genus *Oscillibacter* was detected in human gut microbiota related to some pathologic state [32,33,34]. Elevated *Oscillibacter* abundance had been found in patients with stroke and closely related to gut permeability and host inflammation [34,35]. However, the physiological role of *Oscillibacter* with respect to kidney disease has not been reported. *Veillonella*, another core microbiota associated with CKD stages, has not been studied in gut microbiota from patients with kidney disease, although they were previously reported to be decreased in saliva and oral swab samples of patients with CKD [36].

In the present study, we conducted predicted functional analyses to detect clues indicating the physiological role of the microbiota in the production of toxins associated with kidney dysfunction. We found that as kidney function deteriorated, there was an increase in the microbial contributions to pyruvate metabolism, particularly with regards to the proportional contribution of *Oscillibacter*-related genes. Moreover, we observed that the elevated abundance of *Oscillibacter* and their contribution to pyruvate metabolism were weighted toward anaerobic glycolysis (represented by lactate production) rather than to aerobic glycolysis as kidney function deteriorated. A recent metabolomics study revealed that citric acid metabolism in the tricarboxylic acid cycle, a subsequent pathway of pyruvate, is the most altered metabolic pathway in patients with nondiabetic CKD stages 3–4 compared to in healthy controls [37]. Furthermore, the genus *Oscillibacter* is significantly associated with serum lactate levels according to an animal study and the genus *Veillonella* is related to the metabolism of lactate to propionate in athletes [38,39]. The changes in lactate metabolism mediated by these microbes promote changes in intestinal pH, a decrease in which has previously been shown to disrupt the growth of certain members of the gut microbiota, thereby modifying microbial and metabolic interactions [40]. Based on these processes, changes in the gut microbiota may influence the production and absorption of uremic metabolites by gut intraluminal environmental disturbance, and vice versa. Further comprehensive studies of the association between changes in local and systemic metabolic pathways and specific microbes in patients with renal disease are needed.

Although the present study has notable strengths, such as including an entire spectrum of CKD stages, there were also limitations. First, we did not establish any clear causal relationships among microbiota, microbial metabolites, and CKD. Although we demonstrated associations between microbes and specific metabolites within the microbiota, the causal mechanisms should be determined in further studies. Second, the number of participants registered in each group was relatively small. However, compared with previous studies of CKD and microbiomes, we evaluated a similar or larger number of participants. Furthermore, although we obtained comparative values at certain times during CKD progression, we did not observe serial changes in renal function over time. Finally, we did not assess interactions between the human host and observed microbiota as immunological factors or perform gene expression analyses. However, this study provides a possible association between gut microbiota and renal functions.

## 5. Conclusions

In this study, we detected significant correlations among the gut microbiota, uremic metabolites, and renal functions in patients with CKD, providing insights into the role of the gut microbiome in the progression of kidney disease. Our findings indicate that the deterioration in renal function observed in patients with CKD is related to increases in the relative abundance of *Oscillibacter* within the gut microbiota, the bacteria of which interact with other gut microbes. Based on our observations, *Oscillibacter* may create a favorable environment for the production of various uremic metabolites, and such action may be related to the altered contribution of *Oscillibacter* to the pyruvate metabolism pathway. Further studies are needed to confirm our results and perform more in-depth analyses of the various interrelationships among the gut microbiome, microbial metabolites, and CKD progression.

## Figures and Tables

**Figure 1 microorganisms-08-00907-f001:**
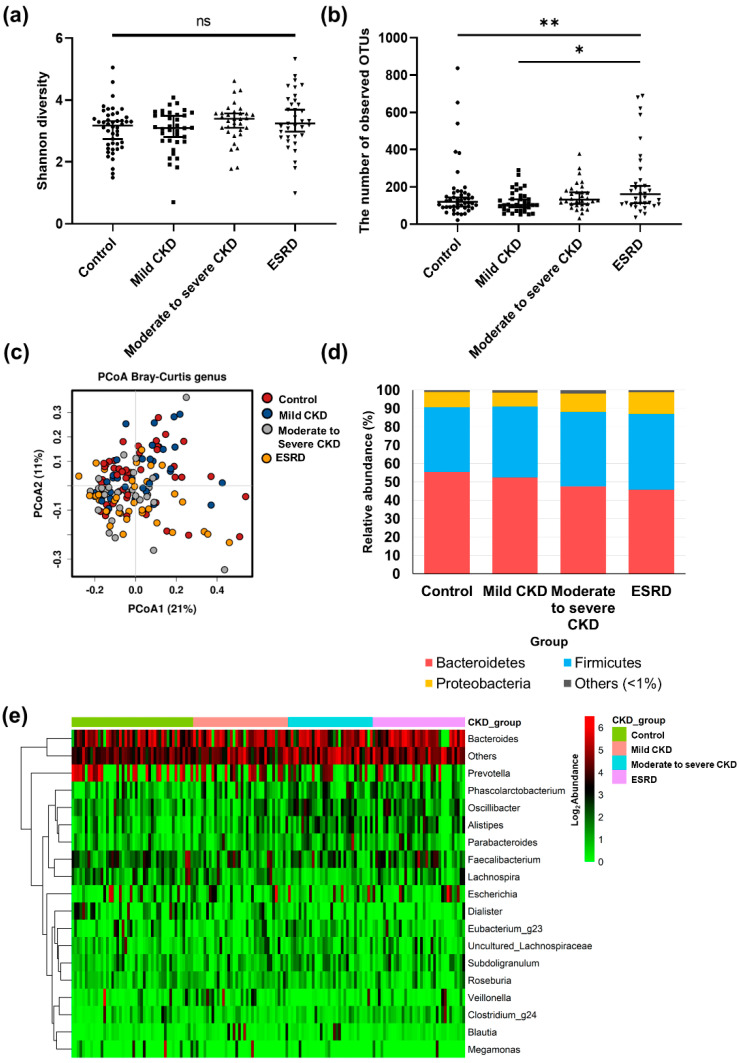
Comparison of diversity and taxonomy of gut microbiota according to chronic kidney disease (CKD) severity. (**a**) Comparison of Shannon diversity indices of gut microbiota among CKD groups. (**b**) Comparison of number of observed operational taxonomic units (OTUs) of gut microbiota among CKD groups (*, *p* < 0.05; **, *p* < 0.01). (**c**) Principal coordinates analysis (PCoA) based on Bray–Curtis distances of gut microbiota among healthy controls (red), patients with mild CKD (blue), patients with moderate CKD (gray), and patients with end-stage renal disease (ESRD, yellow). The first two axes of the PCoA plot are represented by principal coordinate axis 1 (PCoA1) and principal coordinate axis 2 (PCoA2). (**d**) Comparison of microbiota composition among CKD groups at the phylum level. red color, *Bacteroidetes*; blue color, *Firmicutes*; yellow color, *Proteobacteria*; gray color, and other phyla with mean relative abundances <1% of total abundance in samples. (**e**) Comparison of microbiota composition among CKD groups at the genus level. The heatmap plot shows mainly detected genera with mean relative abundances >1% of total abundance in samples; values <1%, unclassified or unidentified, are classified as “others.” The abundances of all genera were plotted after conversion to a binary logarithmic scale.

**Figure 2 microorganisms-08-00907-f002:**
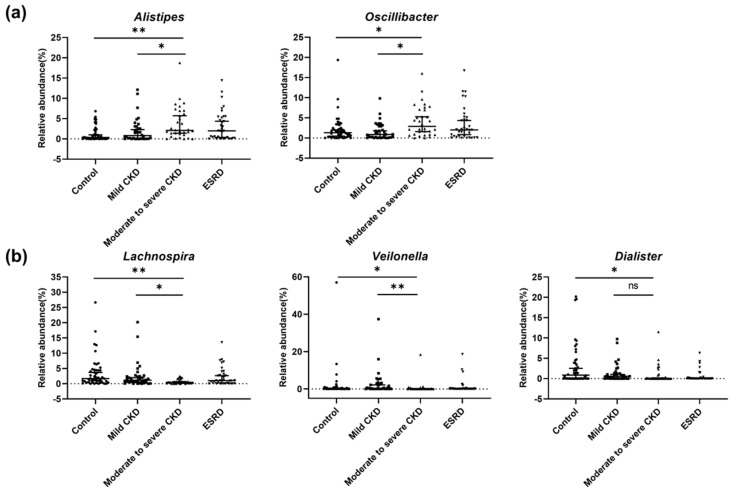
Significantly different genera according to renal function in chronic kidney disease (CKD) patients. We determined bacterial genera showing significant differences among patient groups. (**a**) Genera showing significant increasing trends according to CKD group. (**b**) Genera showing significant decreasing trends according to CKD group. *q*-values were determined using the Benjamini–Hochberg method based on the *p*-values obtained using Mann–Whitney U tests. Single and double asterisks for group comparisons indicate *q* < 0.05 and *q* < 0.005, respectively. ns, not significant.

**Figure 3 microorganisms-08-00907-f003:**
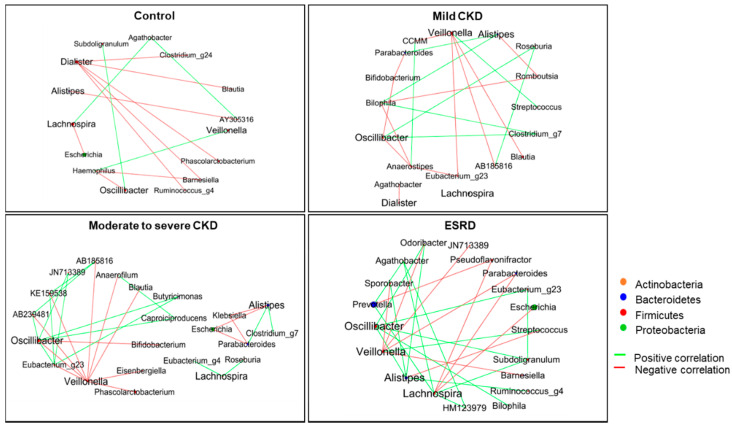
Co-occurrence network for significantly different genera associated with different chronic kidney disease (CKD) groups. The network shows only significant correlations with *q* < 0.05. Green lines denote a positive correlation between microbiota, whereas red lines denote a negative correlation. The size of circles indicates the relative abundance of each genus.

**Figure 4 microorganisms-08-00907-f004:**
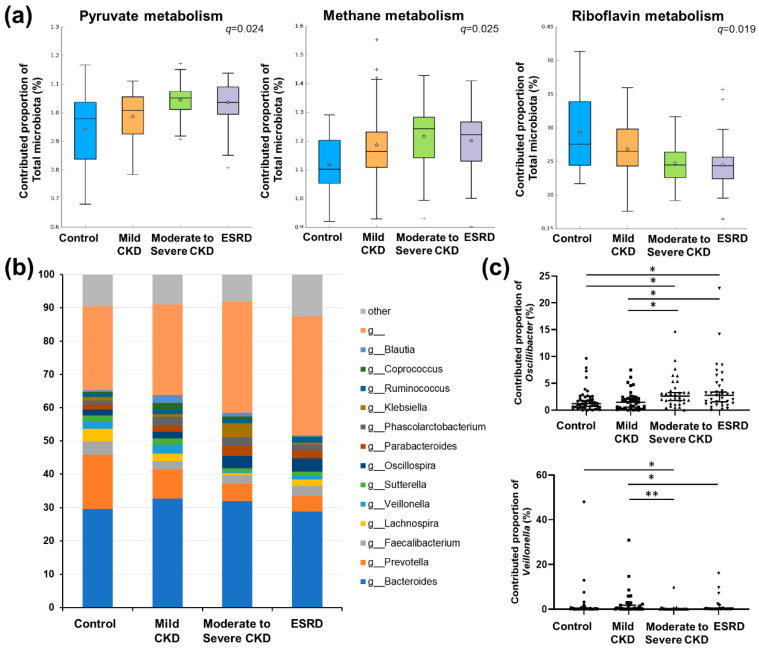
Predicted functional analysis. (**a**) Comparison of significantly different metabolic pathways among chronic kidney disease (CKD) groups based on predicted functional analysis. Only pathways showing a significant difference among groups were plotted (see Appendix A for a complete list of functional pathways). Single and double asterisks for group comparisons indicate *q* < 0.05 and *q* < 0.005, respectively. (**b**) Comparison of the total contribution of genera to the pyruvate metabolism pathway among groups. Only major genera showing a mean proportional contribution >1% among all samples are listed. (**c**) Comparison of the proportional contributions of *Oscillibacter* (upper) and *Veillonella* to the pyruvate metabolism pathway among patient groups. *q*-values were determined using the Benjamini–Hochberg method based on the *p*-values obtained using Mann–Whitney U tests. *, *q* < 0.05; **, *q* < 0.005.

**Figure 5 microorganisms-08-00907-f005:**
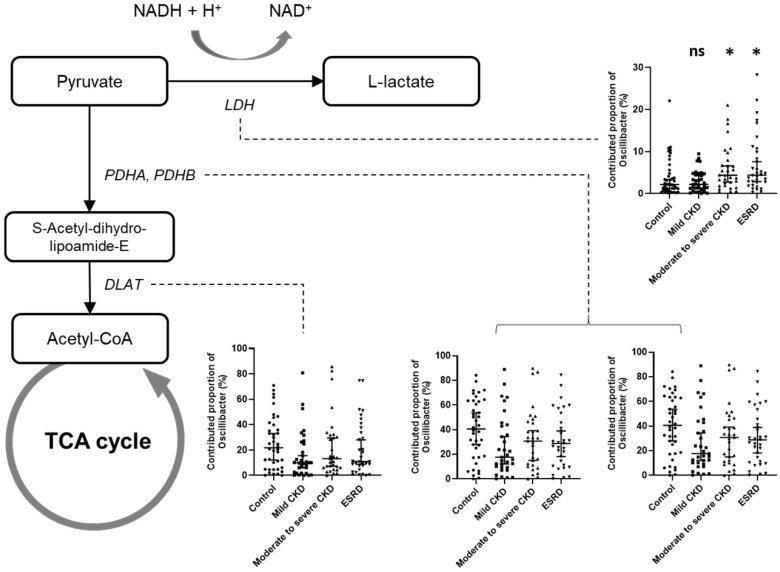
Schematic diagram showing the contribution of *Oscillibacter* to pyruvate metabolism according to renal function. Flow charts were derived based on the Kyoto Encyclopedia of Genes and Genomes (KEGG) pathways. Small bar charts represent a comparison of the contribution of *Oscillibacter* to different orthologous genes according to the patient group. *q*-values were determined using the Benjamini–Hochberg method based on the *p*-values obtained using Mann–Whitney U tests. Asterisks in the bar charts indicate significance (*q*-value) compared with the control group. ns, not significant; *, *q* < 0.05.

**Table 1 microorganisms-08-00907-t001:** Baseline characteristics between groups according to chronic kidney disease (CKD) severity.

Variables	Control	Mild CKD	Moderate to Severe CKD	ESRD	
Total N = 149	N = 46	N = 36	N = 32	N = 35	*p*
Clinical parameters					
Age (y)	47.0 ± 10.8	49.8 ± 15.1	52.4 ± 11.9	48.9 ± 12.2	0.251
Male sex (%)	16 (34.8)	21 (58.3)	17 (53.1)	15 (42.9)	0.15
Body mass index (%)	23.3 ± 3.0	24.6 ± 3.5	23.6 ± 3.2	21.8 ± 4.3	0.003
Diabetes mellitus (%)	0 (0)	3 (8.3)	6 (18.8)	6 (17.1)	0.02
Hypertension (%)	3 (6.5)	21 (58.3)	22 (68.8)	26 (74.3)	<0.001
Blood urea nitrogen (mg/dL)	12.1 ± 2.9	14.4 ± 3.9	43.7 ± 25.7	45.8 ± 15.8	<0.001
Serum creatinine (mg/dL)	0.7 ± 0.2	0.8 ± 0.2	3.8 ± 2.5	7.8 ± 2.6	<0.001
CKD-EPI eGFR (mL/min/1.73 m^2^)	101.6 ± 19.0	98.3 ± 26.1	25.2 ± 17.2	7.2 ± 2.5	<0.001
Urine RBC (number /HPF)					<0.001
0	24 (52.2)	4 (11.1)	7 (21.9)	NA	
1–4	17 (37.0)	7 (19.4)	9 (28.1)	NA	
5≤	5 (10.9)	25 (69.4)	16 (50.0)	NA	
Urine protein/creatinine ratio	0.05 ± 0.03	3.6 ± 3.4	3.3 ± 3.4	NA	<0.001
Plasma hemoglobin (g/dL)	13.8 ± 1.3	12.9 ± 1.6	11.1 ± 2.0	10.3 ± 1.6	<0.001
Anemia (%)	3 (6.5)	13 (36.1)	24 (75.0)	30 (85.7)	<0.001
Serum albumin (mg/dL)	4.4 ± 0.3	3.5 ± 0.7	3.8 ± 0.5	3.8 ± 0.4	<0.001
Serum C-reactive protein (mg/dL)	0.1 ± 0.4	0.2 ± 0.3	0.7 ± 1.3	0.3 ± 0.8	0.321
Etiology of CKD (biopsy proven/clinical diagnosis)					<0.001
Diabetes mellitus	NA	0	4 (0/4)	6 (0/6)	
Hypertension	NA	0	1 (1/0)	1 (0/1)	
Glomerulonephritis	NA	35 (35/0)	21 (18/3)	14 (4/10)	
Polycystic kidney	NA	0	4 (0/4)	3 (0/3)	
Others	NA	1	2	11	
Serum uremic metabolites					
P-cresyl sulfate (ug/mL)	9.5 ± 10.8	7.00 ± 8.7	63.2 ± 56.0	111.6 ± 87.0	<0.001
P-cresyl glucuronide * (ng/mL)	18.2 ± 18.0	19.8 ± 19.3	114.5 ± 110.1	746.7 ± 880.5	<0.001
Indoxyl sulfate (ug/mL)	0.7 ± 0.4	0.7 ± 0.6	7.3 ± 7.6	26.0 ± 17.8	<0.001
TMAO (ug/mL)	0.6 ± 1.1	0.8 ± 1.2	4.9 ± 5.9	13.9 ± 17.4	<0.001

* Calculated excluding samples measured below the minimum measurement limit of *p*-cresyl glucuronide (5 ng/mL). Abbreviations: CKD, chronic kidney disease; eGFR, estimated glomerular filtration rate; RBC, red blood cell; HPF, high-power field; TMAO, trimethylamine N-oxide; NA, not available.

**Table 2 microorganisms-08-00907-t002:** Linear regression with each metabolite and major genera.

Predictors	Regression Coefficient	Standard Error	Adjusted *R*^2^	*p*	FDR
***p*-cresyl sulfate (log)**					
*Alistipes*	0.207	0.050	0.100	<0.001	<0.001
*Oscillibacter*	0.238	0.045	0.155	<0.001	<0.001
*Lachnospira*	–0.105	0.043	0.033	0.016	0.039
*Veillonella*	–0.079	0.027	0.050	0.004	0.014
*Subdoligranulum*	0.234	0.085	0.042	0.007	0.023
*Megamonas*	–0.092	0.036	0.036	0.012	0.034
***p*-cresyl glucuronate (log) ***					
*Prevotella*	–0.023	0.009	0.034	0.014	0.062
*Alistipes*	0.189	0.057	0.062	0.001	0.010
*Oscillibacter*	0.213	0.053	0.094	< 0.001	0.001
*Lachnospira*	–0.104	0.049	0.024	0.034	0.103
*Subdoligranulum*	0.224	0.097	0.028	0.023	0.081
**Indoxyl sulfate (log)**					
*Alistipes*	0.126	0.043	0.048	0.004	0.035
*Oscillibacter*	0.112	0.041	0.043	0.007	0.037
*Lachnospira*	–0.091	0.036	0.034	0.014	0.058
*Subdoligranulum*	0.177	0.073	0.032	0.016	0.058
**TMAO (log)**					
*Prevotella*	–0.016	0.007	0.027	0.026	0.080
*Alistipes*	0.104	0.046	0.027	0.026	0.080
*Oscillibacter*	0.145	0.042	0.067	0.001	0.006
*Lachnospira*	–0.087	0.039	0.026	0.027	0.080
*Dialister*	–0.107	0.043	0.034	0.014	0.080

Only variables with *p*-values under 0.05 are listed in this table while FDR was calculated with all major genera. Listed variables are in descending order according to relative abundance. * Samples measured below the minimum measurement limit (5 ng/mL) are calculated as imputation to zero. Abbreviations: FDR, false discovery rate; TMAO, trimethylamine N-oxide.

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
