# Peer review of "The Association between Gut Microbiota and Uremia of Chronic Kidney Disease"

_microorganisms, 2020, doi:10.3390/microorganisms8060907_

Round 1

Reviewer 1 Report

Chronic kidney diseases (CKDs) display a severe health burden and were associated with numerous co-morbidities which have an additional impact on the patients’ quality of life. Despite the fact that the pathophysiological relevance of microbial dysbiosis was implicated in inflammatory bowel diseases in previous studies, the association between CKD burden and gut microbiota is poorly understood. In their study, the authors aimed to elucidate the microbiota composition in patients suffering from CKDs. In the proposed manuscript, the authors provide clear evidence that the gut microbiome affects the progression of CKDs and that especially Oscillibacter might be involved. This study will be of interest to the readership of Microorganisms, however, a number of minor points should be addressed prior to publication as detailed below.

Please remove p-values from figure 2. Asterisks are sufficient and it could cause confusion if two asterisks but only one p-value is shown per graph.

Font size should be increased in Figure 3.

Statistics in Figure 4C are not clear (lines without any asterisks).

The style of graphs shown in Figure 5 is completely different from all other graphs. Please be consistent.

Reviewer 2 Report

The manuscript deals with a potent issue of the crucial relationship between gut dysbiosis with emphasis on Oscillibacter  and 3 other microbial population and CKD. The manuscript is well written with an uniform thought flow and supported with apt experimentation. The experiments and parameters used are very sound. There are a few word choice modifications, grammar, scientific jargon usage  that would improve the impact of the manuscript. 

  1. Line#44- please rephrase the sentence to eliminate word choice error.
  2. Line #47-48: needs definite article.
  3. Line #49- please replace secretion with influx or inflow or any other appropriate word.
  4. Line #51- please rephrase
  5. Line #55-57- please rephrase to remove repetition of words.
  6. Line #57- please rephrase also needs reference
  7. Line #84-gut microbiota is radically influenced by antibiotic usage adding a few points with respect to the status of antibiotic usage will help a lot.
